# Modified Variable Step-Size Incremental Conductance MPPT Technique for Photovoltaic Systems

**Isaac Owusu-Nyarko** [1] , **Mohamed A. Elgenedy** [2] , **Ibrahim Abdelsalam** [3] **and Khaled H. Ahmed** [1,*]

1    Department of Electronic and Electrical Engineering, University of Strathclyde, Glasgow G1 1XQ, UK; isaac.owusu-nyarko@strath.ac.uk

2    School of Computing, Engineering and the Built Environment, Glasgow Caledonian University, Glasgow G4 0BA, UK; mohamed.elgenedy@gcu.ac.uk

3    Electrical Engineering Department, Arab Academy for Science and Technology and Maritime Transport (AASTMT), Cairo 2033, Egypt; i.abdelsalam@aast.edu

*    Correspondence: khaled.ahmed@strath.ac.uk; Tel.: +44-0141-548-2516

**Abstract:** A highly efficient photovoltaic (PV) system requires a maximum power point tracker to extract peak power from PV modules. The conventional variable step-size incremental conductance (INC) maximum power point tracking (MPPT) technique has two main drawbacks. First, it uses a pre-set scaling factor, which requires manual tuning under different irradiance levels. Second, it adapts the slope of the PV characteristics curve to vary the step-size, which means any small changes in PV module voltage will significantly increase the overall step-size. Subsequently, it deviates the operating point away from the actual reference. In this paper, a new modified variable step-size INC algorithm is proposed to address the aforementioned problems. The proposed algorithm consists of two parts, namely autonomous scaling factor and slope angle variation algorithm. The autonomous scaling factor continuously adjusts the step-size without using a pre-set constant to control the trade-off between convergence speed and tracking precision. The slope angle variation algorithm mitigates the impact of PV voltage change, especially during variable irradiance conditions to improve the MPPT efficiency. The theoretical investigations of the new technique are carried out while its practicability is confirmed by simulation and experimental results.

**Keywords:** autonomous scaling factor; photovoltaic (PV); slope angle variation; variable step-size INC





## 1. Introduction

Photovoltaic (PV) energy increasingly turns out to be a real promising renewable energy source for generation of electricity. This is due to several benefits such as no noise, cleanness, no fuel cost and slight maintenance requirement. Despite numerous benefits of PV power generation, the high cost of installation and low convention efficiency lead to new challenges for implementation on utility scale [1]. Additionally, the non-linear attributes (the graph of power versus voltage) curve of a PV module, produces an exceptional maximum power point (MPP). Therefore, it poses an additional challenge in PV system operation [2]. The situation is further deteriorated since the characteristics of the PV module depend largely on the level of irradiance and temperature. Considering the high cost of installation and possible low efficiency of the PV module, it is important to ensure that the PV module is operated at MPP to improve its efficiency. The literature shows several suggested MPP tracking (MPPT) algorithms. The algorithms differ in control structure, accuracy, required sensors, response time, cost, simplicity and dynamic behaviour during environmental changes [3]. Based on the control structure, these techniques have been classified into two main groups, namely: direct and indirect techniques [4].

Fractional open-circuit voltage (FOCV) and fractional short-circuit current (FSCC) are typical indirect MPPT techniques [5,6]. These are widely used algorithms due to their

structural simplicity. The FOCV technique uses the approximate constant relationship between voltage (Vmpp) at MPP and open circuit voltage ($V_{OC}$) of PV module to track MPP [7,8], while the FSCC technique tracks MPP by using the linear relationship between short circuit current ($I_{SC}$) and current at maximum power point ($I_{mpp}$) of the PV module instead [9]. The indirect techniques have faster tracking capabilities with a minimum number of sensors; however, they only provide an estimation of MPP values. Additionally, power interruption occurs during the measurement of ($V_{OC}$) or ($I_{SC}$).

On the contrary, the most frequently employed direct MPPT techniques are incremental conductance (INC), perturb and observe (P&O) and hill climbing [10,11]. Hill climbing/P&O algorithms have been applied in PV structures because they are cost effective, simple in structure and straightforward to implement [12]. In the P&O technique, MPP is tracked by perturbing the PV module voltage, then examining its effects on the PV power. If the PV power increased in a particular direction of perturbation, then perturbation continues in that direction by the algorithm, unless the direction of perturbation is altered [13]. Similarly, hill climbing is implemented by changing the converter duty cycle and then examine its effect on the output power of the PV module [14,15]. Regardless of these advantages, they can easily lead to wrong judgements, which lead to significant loss of power at MPP under rapidly varying weather conditions [16]. In INC technique, the tracking algorithm uses the slope of the PV module characteristic curve to track MPP. If the slope is zero, that means the operating point is at MPP, positive means it is at the left of MPP and negative means it is at the right of the MPP. Although the INC technique has structural complexity contrary to the P&O and hill climbing algorithms, it can track MPP more accurately with faster response and less steady-state oscillations, thus increasing the tracking efficiency [17–21].

The fixed step-size used in incremental conductance has a significant negative impact on the performance of the PV system under different operating situations. To address this main drawback, a new MPPT technique that uses variable step-size in the INC technique was suggested in [22]. The step-size is controlled as the slope of the PV module characteristic curve varies. A big step-size is used in the INC algorithm if the operating point is initially far from MPP to enhance its tracking speed. While near the MPP, the step-size becomes small to reduce steady-state error. However, the main drawback is that the dynamics of the algorithm are greatly affected by the ratio between the variation of PV power and the variation of PV voltage particularly under variable irradiance. A small variation in PV voltage increases the step-size. This increases the converter duty ratio because the step-size depends on the slope of power and voltage curve of PV module. Thus, the operating point deviates from the actual MPP. Besides, the technique faces another problem, which is the use of a fixed scaling factor. It is very difficult to use a suitable fixed scaling factor for all irradiance levels to ensure zero steady-state error and faster tracking.

Several amendments have been presented to address the drawbacks in variable step-size MPPT technique. In [23], two step-size values were applied to limit oscillations at steady-state. The selected values of step-size only minimise oscillations at the selected irradiance conditions. A new variable step-size fuzzy logic-based INC technique has been discussed in [24]. The fuzzy logic controller varies the fixed step-size value to reach MPP quickly. The algorithm allows fast and accurate convergence under different operating conditions. However, the process is complicated and system dependent, as it is required to store MPP values in advance. In [25], the slope of the PV curve was compared with two arbitrary points on the I-V curve to select three different step-size values suitable for the whole operating range. Although the algorithm improves the dynamic performance, it encounters poor steady-state performance around MPP.

An enhancement was introduced in [26] by removing all division terms to avoid the impact of small changes in the voltage values. Therefore, oscillation is reduced significantly; however, the algorithm experienced slow tracking speed at the initial operating stage of the PV system. On the other hand, modifications have been also presented to select a suitable scaling factor for the conventional variable step-size in INC technique. In [27], a

root locus technique was derived to obtain an optimum value of the scaling factor. The algorithm achieved steady-state accuracy and dynamic response only at selected irradiance conditions. In [28], two different scaling factor values were introduced. Although, the MPP is tracked, selected scaling factors are not optimum under all operating circumstances.

This paper, introduces a modified variable step-size INC technique for MPPT in PV systems. The new MPPT technique controls the step-size without using a constant preset scaling factor. It can exploit all advantages of conventional variable step-size INC technique, while eliminating its drawbacks. This paper is organised into seven sections. Following this introduction, limitations of conventional variable step-size in the INC technique are discussed in Section 2. Section 3 explains the proposed variable step-size algorithm. Simulation results are presented in Section 4. In Section 5, partial shading analysis of the proposed MPPT is presented. In Section 6, experimental set up is presented where the results validate the practicability of the new MPPT technique, while the conclusion is presented in Section 7.

## 2. Conventional Variable Step-Size INC Algorithm

### 2.1. Conventional INC Algorithm

INC MPPT uses the ratio between the PV module power change to the voltage change to track MPP. Using the equivalent circuit model in [29], the output current and output voltage of the PV module can be written in (1) and (2).

$$I_{pv} = I_{ph} - I_o \left[ exp \left( \frac{q(V_{pv} + I_{pv}R_s)}{bT\alpha} \right) - 1 \right] - \frac{(V_{pv} + I_{pv}R_s)}{R_p} \tag{1}$$

$$V_{pv} = \frac{bT\alpha}{q} ln \left( \frac{I_{ph} - I_{pv} + I_o}{I_o} \right) - I_{pv}R_s \tag{2}$$

where, and $R_s$ and $R_p$ are the series and parallel resistances, respectively. $I_o$ and $I_{ph}$ are the reverse saturation and light generated currents, respectively. $T$, $\alpha$, $q$ and $b$ are temperature, ideality factor, charge of electrons and Boltzmann's constant, respectively. The PV module has an output power given as:

$$P_{pv} = V_{pv} I_{pv} \tag{3}$$

By differentiating Equation (3), the slope of PV module is obtained in Equation (4).

$$\frac{dP_{pv}}{dV_{pv}} = \left( I_{pv} + V_{pv} \frac{dI_{pv}}{dV_{pv}} \right) \tag{4}$$

Thus, INC algorithm can be expressed as in (5)–(7).

$$\frac{dP_{pv}}{dV_{pv}} = 0, \ \frac{dI_{pv}}{dV_{pv}} = -\frac{I_{pv}}{V_{pv}} \ \text{at the MPP} \tag{5}$$

$$\frac{dP_{pv}}{dV_{pv}} > 0, \ \frac{dI_{pv}}{dV_{pv}} > \frac{I_{pv}}{V_{pv}} \ \text{at the left of MP} \tag{6}$$

$$\frac{dP_{pv}}{dV_{pv}} < 0, \ \frac{dI_{pv}}{dV_{pv}} < \frac{I_{pv}}{V_{pv}} \ \text{at the left of MP} \tag{7}$$

Therefore, MPP is tracked by comparing $(dI_{pv}/dV_{pv})$ to $(I_{pv}/V_{pv})$. The operation of the PV module is kept around MPP once MPP is reached unless there is a change in the current of PV module. Then, the algorithm controls the PV module voltage to follow the reference to continuously track the new MPP [30].

### 2.2. Conventional Variable Step-Size Algorithm

A conventional variable step-size algorithm controls the duty cycle of the converter by controlling the step-size to reach MPP. This MPPT technique works differently to the

conventional INC algorithm with a pre-set step-size. The generated PV module output power with a large step size contributes to excessive power oscillations at steady-state, but faster dynamics response resulting in a comparatively low tracking efficiency. An opposite scenario occurs when smaller step-size is applied in the MPPT. Thus, without a suitable optimum step-size value, both steady-state accuracy and fast dynamics cannot be achieved simultaneously. Using a variable step-size could address this challenge very well, as the step-size gets smaller when the algorithm approaches MPP. Conventional variable step-size is given in (8) [22].

$$D(k) = D(k-1) \pm N \left| \frac{dP_{pv}}{dV_{pv}} \right| \tag{8}$$

where,

$$dP_{pv} = P_{pv}(k) - P_{pv}(k-1) \tag{9}$$

$$dV_{pv} = V_{pv}(k) - V_{pv}(k-1) \tag{10}$$

where, $D(k)$ and $D(k-1)$ are the present and previous duty cycle at $k$ and $k-1$ samples, respectively, $P_{pv}(k)$ and $P_{pv}(k-1)$ are the present and previous PV module power at $k$ and $k-1$ samples, respectively, and $V_{pv}(k)$ and $V_{pv}(k-1)$ are the voltage at $k$ and $k-1$ samples. $N$ presents the preset scaling factor necessarily used to fine-tune the step-size at the design stage to compromise between dynamic response and steady state precision. Figure 1a shows the behaviour of the conventional variable step-size INC in a typical PV system under different irradiance levels. The PV module output power oscillates around the MPP when PV module operates around its optimum point. It is obvious, as shown in Figure 1a, that the PV module output power slowly moves to the new MPP when irradiance changes. The duty cycle in Figure 1b verifies the converter response under variable irradiance conditions. Significant duty cycle oscillations around optimum value are experienced at steady state. These oscillations are due to the high step-size value during small PV voltage change. Figure 2 verifies the impact of the scaling factor on the PV output power. It can be seen that the large scaling factor in conventional variable step-size compared to the optimum one can provide a faster dynamic response, but excessive oscillations at steady state.

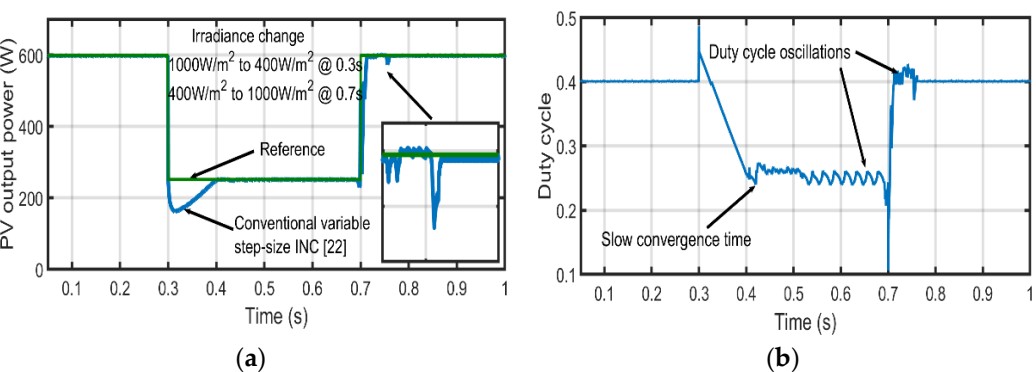

**Figure 1.** Drawbacks of conventional variable step-size INC MPPT with optimum scaling factor: (**a**) PV modules output power, and (**b**) duty cycle.

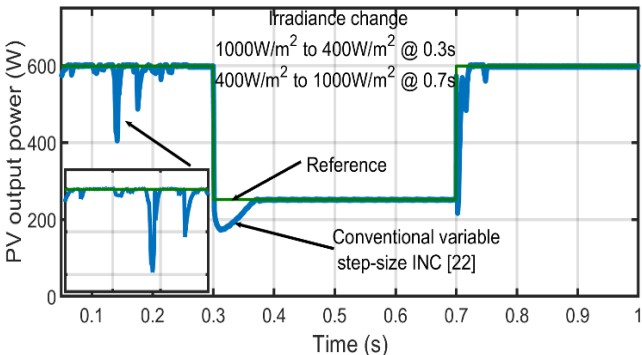

**Figure 2.** PV module output power characteristic with bigger scaling factor compared to optimum value.

## 3. Proposed Variable Step-Size INC Algorithm

The PV module operating voltage ranges between $0-V_{oc}$, however, unnecessary sampling within large range slows down the tracking speed. Limiting the search range restricts the viable operating range, thus reducing the tracking time for the MPPT algorithm. Therefore, an initial sampling value of 76% of open-circuit voltage is embedded in the proposed algorithm to restrict the search range [31]. This is to enable the proposed MPPT to record fewer perturbation directions before converging at MPP. Hence, the structure of the proposed MPPT technique for the PV system is defined by dividing the search range of the P-V characteristic curve shown in Figure 3 into three regions, namely A, B and C. For satisfactory speed response, the operating range for the proposed MPPT technique is given in inequality (11).

$$0.76V_{oc} \leq V_{pv}(k) \leq V_{oc} \tag{11}$$

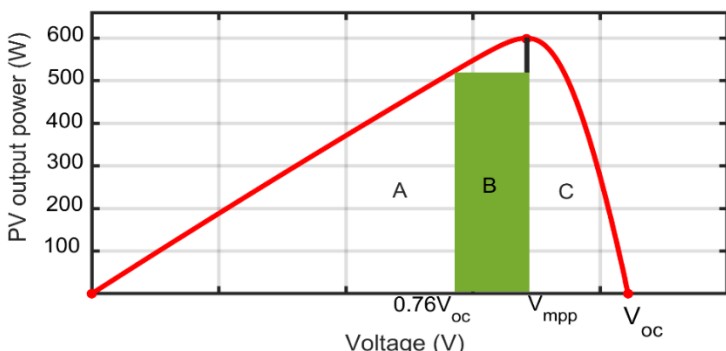

**Figure 3.** P-V characteristic curve.

The proposed MPPT algorithm is divided into three parts and explained as follows.

### 3.1. Proposed Autonomous Scaling Factor

The conventional variable step-size technique includes a pre-set scaling factor, which cannot achieve both fast dynamic response and steady-state precision. Therefore, the scaling factor should be varied to reduce the power loss at steady state. Figure 3 shows the characteristic curve of the PV module and the study of it demonstrates an important observation near the MPP. The output voltage variation of PV module is mimicked to vary the scaling factor to enhance the tracking performance during transient and steady-state situations. The range of voltage change for region B and C are, respectively, given in Equations (12) and (13).

$$\Delta V_B = V_{pv}(k) - 0.76V_{oc} \tag{12}$$

$$\Delta V_C = V_{oc} - V_{pv}(k) \tag{13}$$

To ensure an adjustable scaling factor towards the MPP, a new autonomous scaling factor is proposed, which can be illustrated in terms of $\Delta V_B$ and $\Delta V_C$ given in (14).

$$N_D = \frac{\Delta V_B}{\Delta V_C} = \left| \frac{V_{pv}(k) - 0.76V_{oc}}{V_{oc} - V_{pv}(k)} \right| \tag{14}$$

### 3.2. Estimation of $V_{oc}$

For accurate operation plus avoid adding any extra hardware components, the proposed variable MPPT technique estimates the $V_{oc}$ value. This voltage is required in calculating the proposed autonomous scaling factor. Most of MPPT algorithms use either temperature or irradiance sensors or both to estimate $V_{oc}$. Therefore, the value can be estimated given in Equation (15) as in [32].

$$V_{oc} = V_{oc\_STC} + \alpha V_t In \left( \frac{G}{G_{STC}} \right) + K_v(T - T_{STC}) \tag{15}$$

where, $G_{STC}$, $V_{oc\_STC}$, and $T_{STC}$ are the solar irradiance, open circuit voltage and temperature at standard test conditions, respectively. $G$, $V_t$, $T$ and $K_v$ are the operating solar irradiance, thermal voltage, operating temperature, and temperature coefficient of $V_{oc}$, respectively [33]. The output current of the PV module and its short circuit current, $I_{SC\_STC}$ has a linear relationship [34] as in (16).

$$I_{PV} = (I_{SC\_STC} + K_I (T - T_{STC})) \frac{G}{G_{STC}} \tag{16}$$

Hence, the estimation of $V_{oc}$ for the PV module given in (15) is updated using Equation (16) as follows:

$$V_{oc} = V_{oc\_STC} + \alpha V_t In \left( \frac{I_{pv}}{I_{SC\_STC}} \right) \tag{17}$$

The estimated value of the open-circuit voltage for different irradiance levels is summarized in Table 1. The estimated values are close to the actual values. Although, under lower irradiance levels as in Figure 4, the expression in (17) shows deviations from the actual value, but the deviation is below 3%, which is substantially low and satisfactory for a typical PV module system.

**Table 1.** Estimation of the module open circuit voltage.

| Irradiance (W/m²) | 1000 | 800 | 600 | 400 | 200 |
|---|---|---|---|---|---|
| Ambient current (A) | 3.500 | 2.826 | 2.11 | 1.421 | 0.7099 |
| Actual $V_{oc\_array}$ | 211.00 | 209.60 | 207.70 | 205.30 | 201.70 |
| Estimated $V_{oc\_array}$ | 210.21 | 208.18 | 205.38 | 201.56 | 195.89 |
| Deviation | 0.37% | 0.68% | 1.11% | 1.82% | 2.88% |

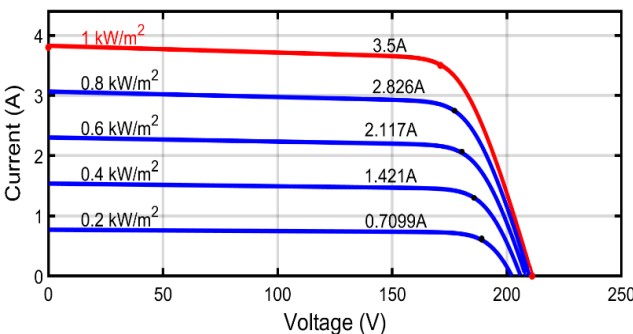

**Figure 4.** Voltage—current characteristic curve for PV modules.

### 3.3. Slope Angle Variation Algorithm

The conventional variable step-size presented in (8) is dependent on the slope of the P-V characteristic curve; hence, it exhibits dynamic performance deterioration under varying irradiance conditions. There is a notable change of power (changing in MPP) when the irradiance level changes from $G_1$ to $G_2$ as shown in Figure 5, while there is a relatively small change in PV voltage. This results in a significantly large step-size value, which may push the MPPT algorithm to take more time to reach the new MPP. To address this issue, the angle between PV power change and related voltage change is utilized. This can control the change in power, which eventually controls the step-size value irrespective of the variation of PV voltage. When the angle is small, the PV power change also becomes small to limit the large increase in step size. The slope angle variation can be derived from PV output power as follows:

$$\frac{dP_{pv}}{dV_{pv}} = \frac{\sin \delta}{\cos \delta} \tag{18}$$

then,

$$\left| \frac{\sin \delta}{\cos \delta} \right| = \left| I_{pv} + V_{pv} \frac{dI_{pv}}{dV_{pv}} \right| \tag{19}$$

where, $\delta$ is the angle between the PV module output power variations to the voltage variation.

$$\sin \delta = \left( I_{pv} + V_{pv} \frac{dI_{pv}}{dV_{pv}} \right) \cos \delta \tag{20}$$

$$\sin \delta = \left( I_{pv} + V_{pv} \frac{dI_{pv}}{dV_{pv}} \right) \frac{dV_{pv}}{\sqrt{dP_{pv}^2 + dV_{pv}^2}} \tag{21}$$

$$\sin \delta = \frac{I_{pv} dV_{pv}}{\sqrt{dP_{pv}^2 + dV_{pv}^2}} + \frac{V_{pv} dI_{pv}}{\sqrt{dP_{pv}^2 + dV_{pv}^2}} \tag{22}$$

$$\sin \delta = \frac{dP_{pv}}{\sqrt{dP_{pv}^2 + dV_{pv}^2}} \tag{23}$$

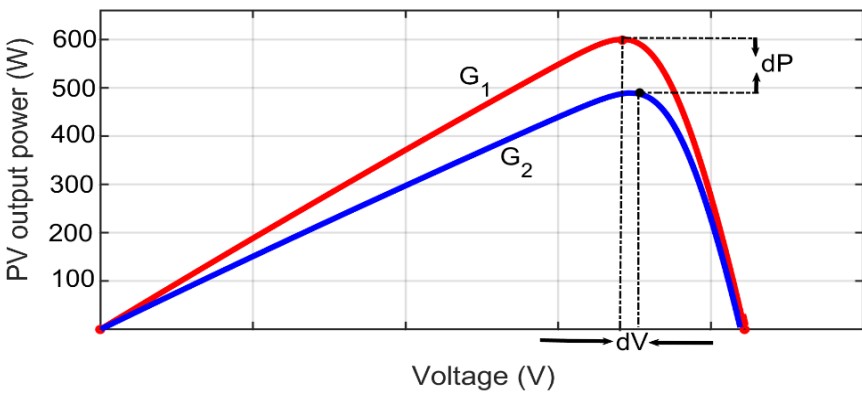

**Figure 5.** Effect of irradiance on MPP.

For a small value of $\delta$,

$$\sin\delta = \tan \delta = \frac{\sin \delta}{\cos \delta} = \frac{dP_{pv}}{dV_{pv}} \tag{24}$$

Using Equations (14) and (23), the proposed duty cycle using the new slope angle variation algorithm can be calculated as:

$$D(k) = D(k-1) \pm \left(\frac{\Delta V_B}{\Delta V_C}\right) \frac{dP_{pv}}{\sqrt{dP_{pv}^2 + dV_{pv}^2}} \tag{25}$$

Hence, the new proposed variable step-size is given by:

$$\Delta D_{mod} = \left(\frac{\Delta V_B}{\Delta V_C}\right) \frac{dP_{pv}}{\sqrt{dP_{pv}^2 + dV_{pv}^2}} \tag{26}$$

Figure 6 shows the flow chart of the proposed MPPT technique.

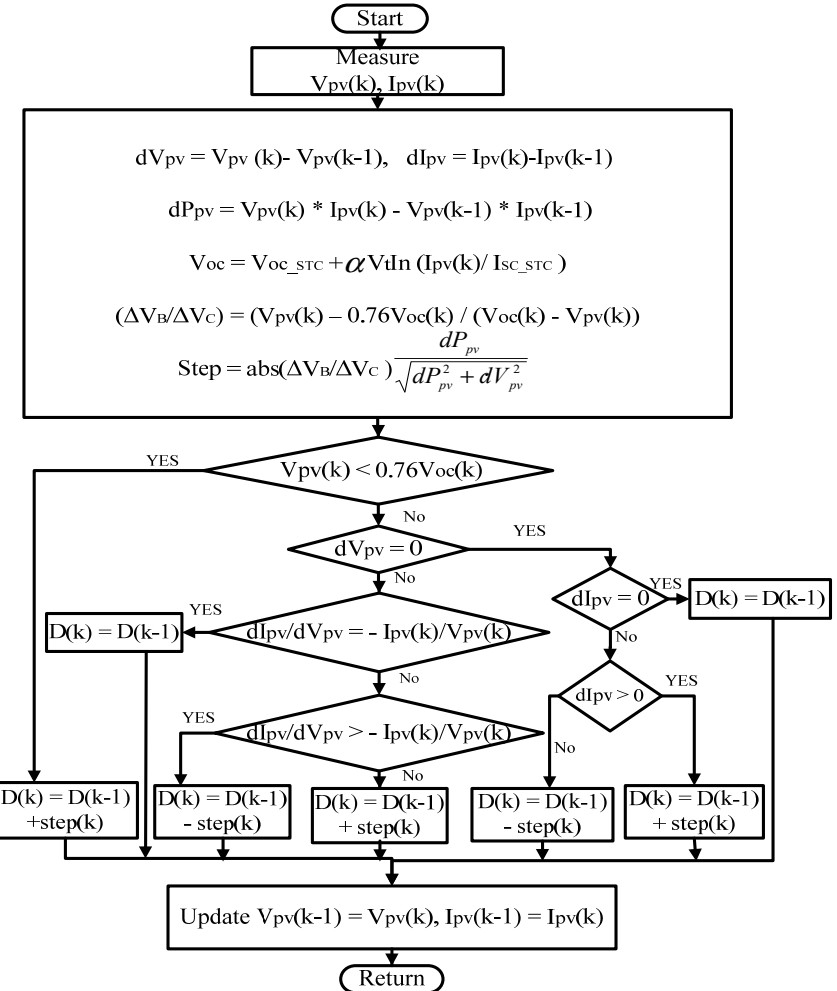

**Figure 6.** Flow chart of the proposed variable step-size INC MPPT.

## 4. Simulation Results

The PV system under test consists of ten series-connected PV modules (the whole rated power is 600 W), a dc battery, and a dc-dc boost converter. The tested PV system is shown in Figure 7 and it is grid connected PV system, where the battery represents the grid and interfacing inverter. Table 2 shows the PV module specifications. The simulation is carried out under different irradiance levels. The initial irradiance is 1000 W/m$^2$ and is changed to 400 W/m$^2$ at 0.3 s. The irradiance level then changed back to 1000 W/m$^2$ at 0.7 s. The proposed MPPT technique, the conventional variable step-size INC [22] and two existing modified techniques in [26,28] are simulated and compared at same conditions

of irradiance and temperature. Figure 8a is the output power of the PV modules under variable irradiance conditions for the proposed and conventional variable step-size INC MPPT techniques. It proves that the proposed technique is more effective than conventional MPPT technique. The proposed algorithm tracks the MPP faster with minimum oscillations around MPP compared with the conventional counterpart. Figures 8b and 9a represent the converter duty cycle of proposed and conventional variable step-size INC MPPT, respectively. The duty cycle of proposed variable step-size INC MPPT reaches its optimum value with small oscillations within a satisfactory time. Furthermore, during the changes in irradiance, the duty cycle of the proposed technique moves toward the optimum value faster. On the contrary, the converter duty cycle of conventional MPPT technique shown in Figure 8a exhibits significant oscillations around its optimum value. The performance of the new autonomous scaling factor is shown in Figure 9b. The scaling factor is adjusted respective to irradiance change to further improve the dynamic response. Thus, with an embedded autonomous scaling factor, the proposed MPPT tracking time is faster in start-up and for a large change in irradiance compared to conventional variable step-size INC technique with a fixed scaling factor. The proposed algorithm shows a tremendous reduction of power oscillations at both dynamic and steady-state, and MPP is directly reached under changes in irradiance conditions. To further test the robustness of the proposed algorithm, Figure 10 compares the proposed technique and two other existing modified variable step-size INC MPPTs. All algorithms track optimum power point, however, the proposed MPPT technique is faster than other two techniques in [26,28]. The variable step-size INC technique in [26] tracks better than [28] under step change in irradiance. However, due to the elimination of the division terms (change in PV module voltage), which eventually decreases the step-size, the algorithm experiences slow tracking at start up compared with the proposed MPPT technique. Additionally, scaling factors in [26,28] are not optimum under all operating conditions. This means that under fast changing weather conditions, both techniques could fail to track MPP. Tables 3–5 present a performance comparison between the proposed MPPT technique, the conventional variable step-size INC [22] and two existing modified techniques in [26,28]. These tables demonstrate the tracking time, power of oscillations at MPP and tracking efficiency. Further tests were also carried out to ascertain the robustness of the proposed MPPT technique and the three MPPT techniques under investigation using different irradiance levels. The irradiance levels under this scenario are from 800 W/m$^2$ to 400 W/m$^2$ at 0.3 s and then changed back to 400 W/m$^2$ at 0.7 s. Figure 11 shows the output power of the proposed, conventional [22] and existing modified variable step-size MPPT [28] techniques, while Figure 12 depicts the output power of the proposed technique and another existing modified variable step-size MPPT [26] techniques under study. It is evident that the proposed MPPT technique tracks faster as compared to the conventional and the two modified techniques under this condition. Table 6 shows the tracking time of each algorithm under dynamic weather conditions. In general, the proposed algorithm has high dynamic performance, high accuracy, low overshoot and low energy for tracking. Additionally, it is obvious that the proposed MPPT technique provides higher tracking accuracy and does not require a pre-set scaling factor.

**Table 2.** The PV module parameters (MSX60) [33].

| Parameters | Value |
|---|---|
| PV module Short − circuit current ($I_{sc}$) | 3.8 A |
| PV module open − circuit voltage ($V_{oc}$) | 21.1 V |
| Maximum PV module current ($I_{mpp}$) | 3.5 A |
| Maximum PV module Voltage ($V_{mpp}$) | 17.1 V |
| Maximum PV module power ($P_{max}$) | 60 W |
| Temperature coefficient of $V_{oc}$ ($K_v$) | −0.08 V/°C |
| Temperature coefficient of $I_{sc}$ ($K_i$) | 0.005 A/°C |

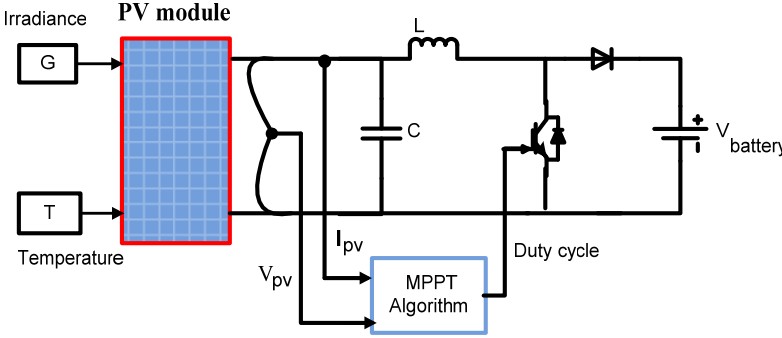

**Figure 7.** PV system under investigation.

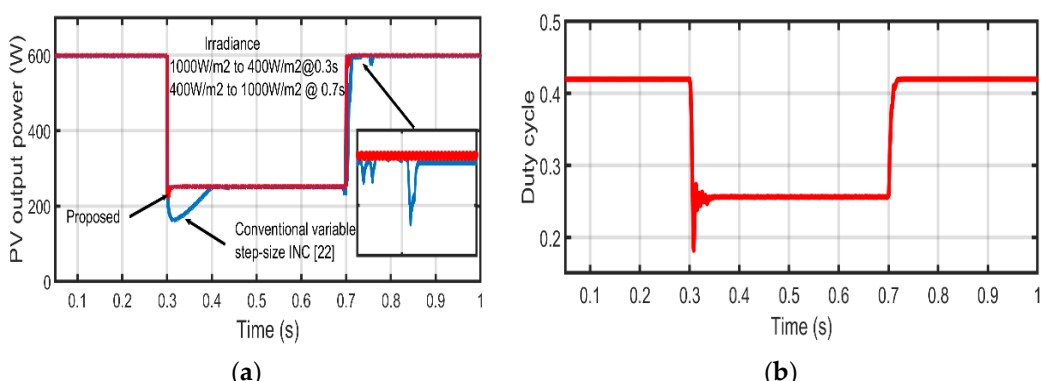

**Figure 8.** Simulation of PV system for conventional and proposed MPPT technique under varying irradiance: (**a**) PV modules output power, (**b**) duty cycle for the proposed technique.

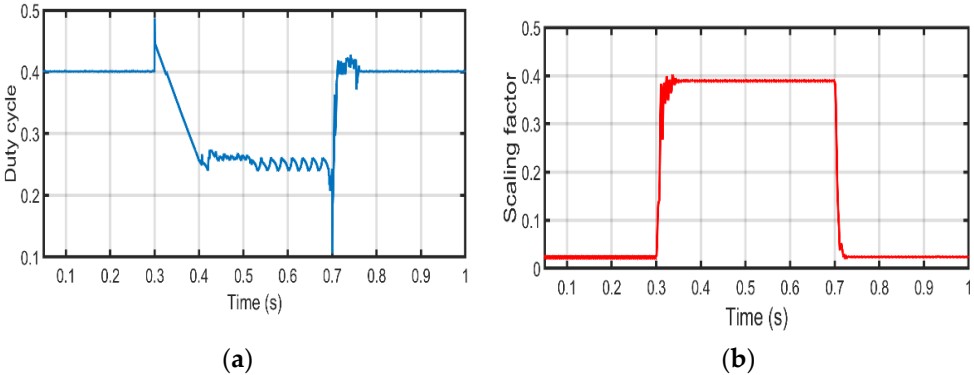

**Figure 9.** Simulation results: (**a**) Duty cycle for conventional variable step-size INC technique; (**b**) proposed autonomous scaling factor during changes in irradiance.

**Table 3.** Performance indicators of the proposed, conventional and the two existing modified variable step-size INC MPPT under variable irradiance.

| Conditions | Variable Step-Size Methods | Energy Used (mJ) | Dynamic Performance | Tracking Accuracy |
|---|---|---|---|---|
| 1000 W/m$^2$ to 400 W/m$^2$ | [22] | 1522 | Low | Low |
| | [26] | 186 | Low | High |
| | [28] | 192 | Low | Low |
| | Proposed | 163 | High | High |

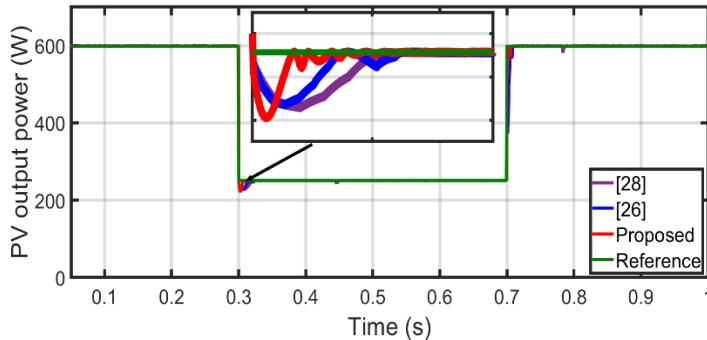

**Figure 10.** The proposed variable step-size MPPT technique and the other two modified MPPT techniques.

**Table 4.** Performance analysis of the proposed, conventional and the two existing modified variable step-size INC MPPT under variable irradiance.

| Conditions | Variable Step-Size INC Methods | Tracking Time (ms) | Efficiency (%) |
|---|---|---|---|
| 1000 W/m² to 400 W/m² | [22] | 120.3 | 98.20 |
| | [26] | 18.4 | 99.18 |
| | [28] | 27.3 | 98.35 |
| | Proposed | 12.6 | 99.84 |

**Table 5.** Tracking comparison of the proposed, conventional and the two existing modified variable step-size INC MPPT under variable irradiance.

| Variable Step-Size Methods | Average Power at 1000 W/m² | Settling Time (s) | Oscillation at MPP, W | Efficiency (%) |
|---|---|---|---|---|
| [22] | 595.2 | 32.88 | 3.3 | 99.45 |
| [26] | 596.4 | 13.17 | 2.1 | 99.65 |
| [28] | 595.7 | 16.50 | 2.8 | 99.53 |
| Proposed | 596.9 | 10.09 | 1.6 | 99.73 |

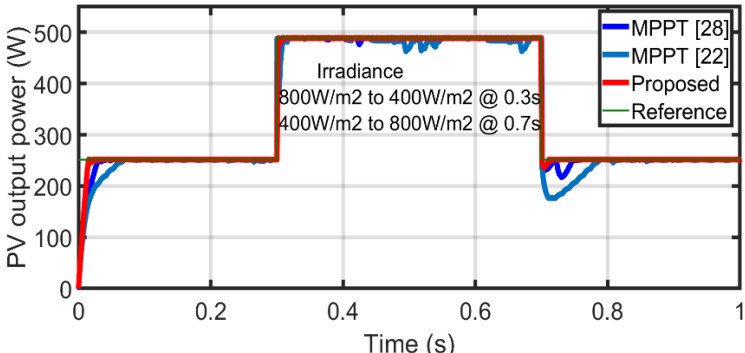

**Figure 11.** Simulation results of the proposed and conventional variable step-size MPPT under variable irradiance conditions.

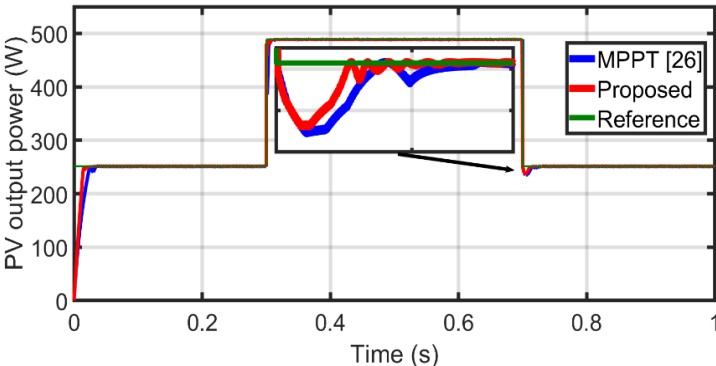

**Figure 12.** Simulation results of the proposed and the two existing modified variable step-size MPPT algorithms under variable irradiance conditions.

**Table 6.** Tracking time with step irradiance change of the proposed, conventional and two existing modified variable step-size INC MPPT techniques.

| Conditions | Variable Step-Size INC Methods | Tracking Time (ms) |
|---|---|---|
| 800 W/m² to 400 W/m² | [22] | 88.8 |
| | [26] | 17.6 |
| | [28] | 31.2 |
| | Proposed | 10.2 |

## 5. Partial Shading Analysis of the Proposed MPPT

Partial shading in the PV system is expected in the PV array where certain portions of the PV modules are exposed to unequally distributed radiation intensity. Under this condition, multiple peaks which consist of local and global maximum power points are generated in the PV characteristic curve. Conventional MPPT fails to track global maximum power point due to lack of resistance of the conventional MPPT to the local maximum power point. The effect is significant power loss in the PV system and waste of energy in the PV system under partial shading condition. The performance of the proposed MPPT technique is tested using scanning technique. Ten PV modules (MSX60) are connected in series with a bypass diode connected in parallel with each PV module. Initially, five PV modules were made to operate at 1000 W/m² while the remaining are partially shaded and operating at 500 W/m². Figure 13 shows the arrangement of the PV modules. The output power of the PV array is shown in Figure 14. Clearly, it evident that the proposed MPPT is resistance to local maxima, as this local maximum power point does not prevent the proposed MPPT from reaching the global maximum power point. To further test the robustness of the proposed MPPT, the shading pattern is changed where three of the PV modules received irradiance of 1000 W/m², four PV modules received 500 W/m² and three of the PV modules received 300 W/m². Figure 15 shows the arrangement of the PV modules with different shading conditions. Figure 16 shows the PV output power, which demonstrates the capability of the proposed MPPT to distinguish global maximum power from the local maximum power. To further clarify and analyse the behaviour of the proposed MPPT, the same partial shading condition of Figure 15 is repeated with different irradiance on unshaded PV modules. The irradiance levels were made to vary from 200 W/m² to 1000 W/m² at 0.5 s and then changed from 1000 W/m² to 200 W/m² at 1 s. Figure 17 shows that the proposed MPPT tracks the global maximum point with short time under variable irradiance conditions. This is due to the limited search space of the proposed algorithm to reach the global maximum power in relatively short time. The wider search space constitutes significant power loss. The ability of the proposed MPPT to track the global maximum power point without preventing from the local maximum power point demonstrates that there is a potential of real energy gain under partial shading

conditions. The global maximum power tracked by the proposed MPPT is about 99.87% with minimum power oscillation.

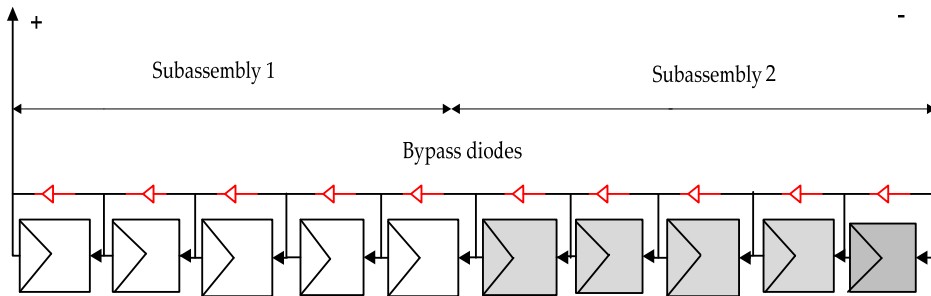

**Figure 13.** PV array under partial shading conditions.

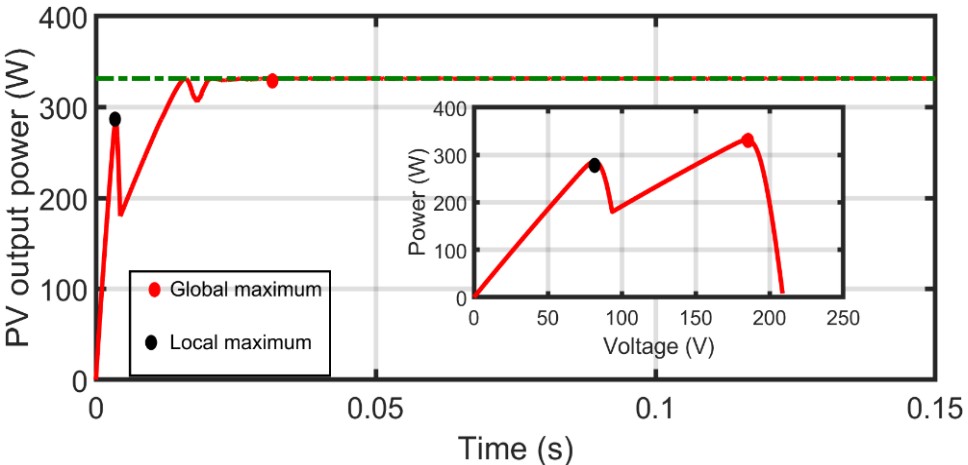

**Figure 14.** Global and local maxima under scanning.

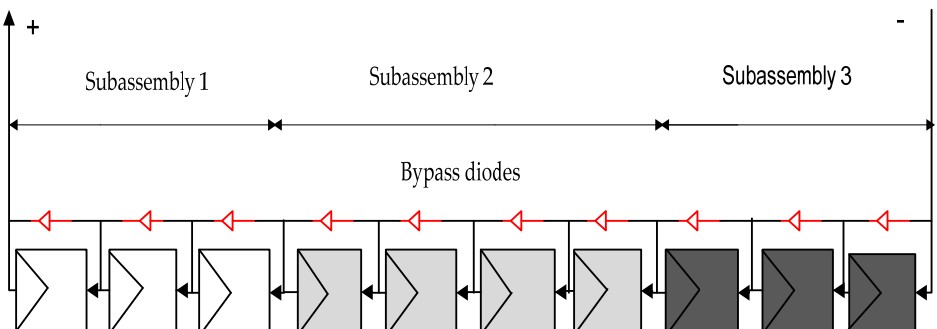

**Figure 15.** PV modules under partial shading condition.

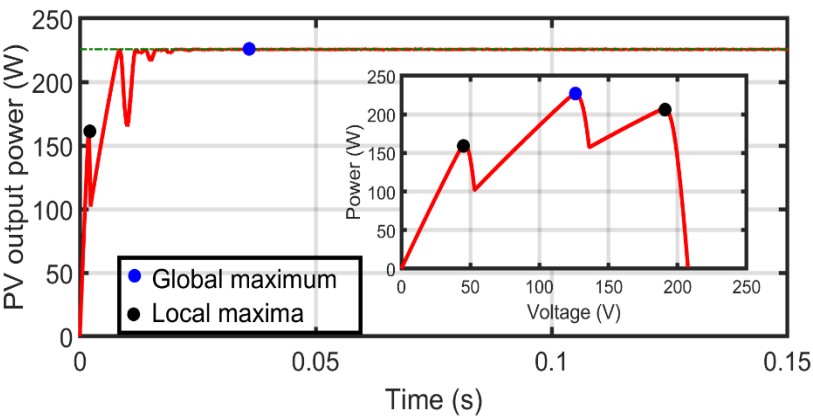

**Figure 16.** PV output power for partially shade PV system.

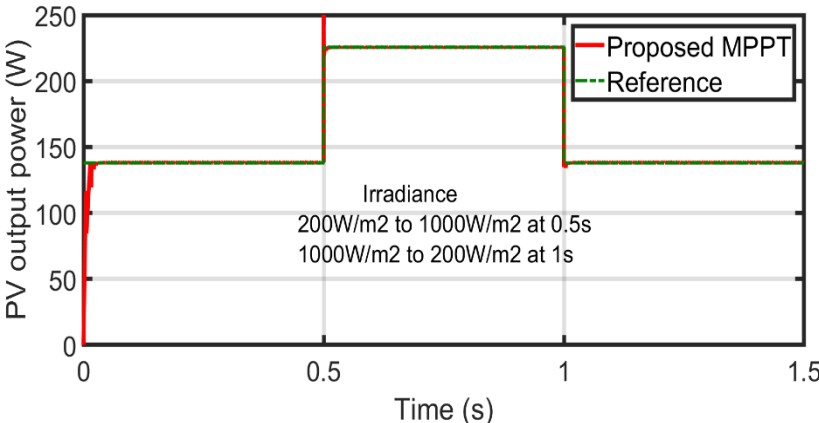

**Figure 17.** Output power for PV arrays under partial shaded condition with two irradiance levels.

## 6. Experimental Results

The proposed MPPT technique and conventional variable step-size INC technique in [22] are experimentally evaluated in this section. Figure 18 shows the experimental prototype of the grid connected PV system. The experimental prototype comprises of boost converter connected to 110 V DC bus (representing the grid and interfacing inverter) and a PV emulator. The experimental setup parameters are listed in Table 7. The PV emulator is based on PV module parameters listed in Table 8, where the maximum output power and the maximum output current are set to the programmable power supply (EA-PS-83600-10 with analogue interface). The microcontroller (CY8C5888LTI-LP097) is used to set the reference output voltage based on the built-in PV model, irradiance, and the output current. For clarity, Figure 19 shows the flow chart of the microcontroller programme for PV emulator. To compare the dynamic performance of the two MPPT algorithms, the irradiance level step change from 1000 W/m² (PV output power is 300 W, PV voltage is 54.7 V) to 500 W/m² (PV output power is 150 W, PV voltage is 51.43 V). The experimental results of the two MPPT techniques are shown in Figure 20. The results show a detail view of the dynamic performance of both the conventional and proposed viable step-size INC techniques during step irradiance change. It can be concluded from the results that the proposed viable step-size INC technique successes to track the maximum power in 60 ms, which is faster than the conventional algorithm that needs around 140 ms to track the MPP. Additionally, it can be observed from Figure 20 parts c and d that the dc-bus delivered output power and is more stable with minimum oscillations at steady state, however, the dc-bus output power is less than the PV rated power at maximum point; due to the boost converter switching and conduction losses and boost converter inductor wire resistance.

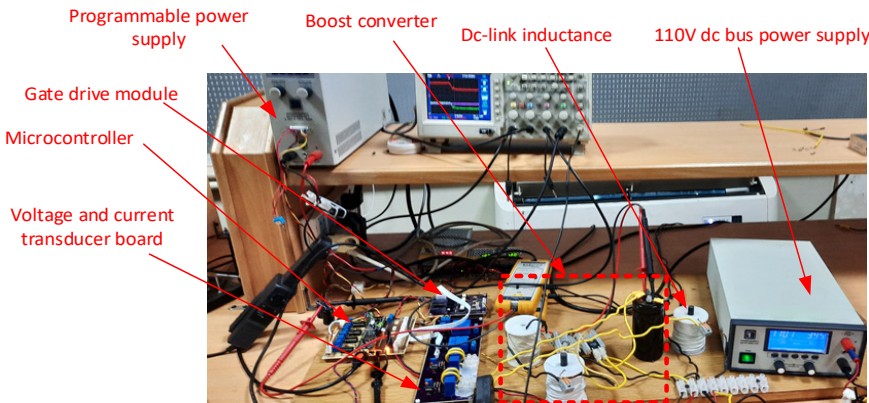

**Figure 18.** Experimental setup.

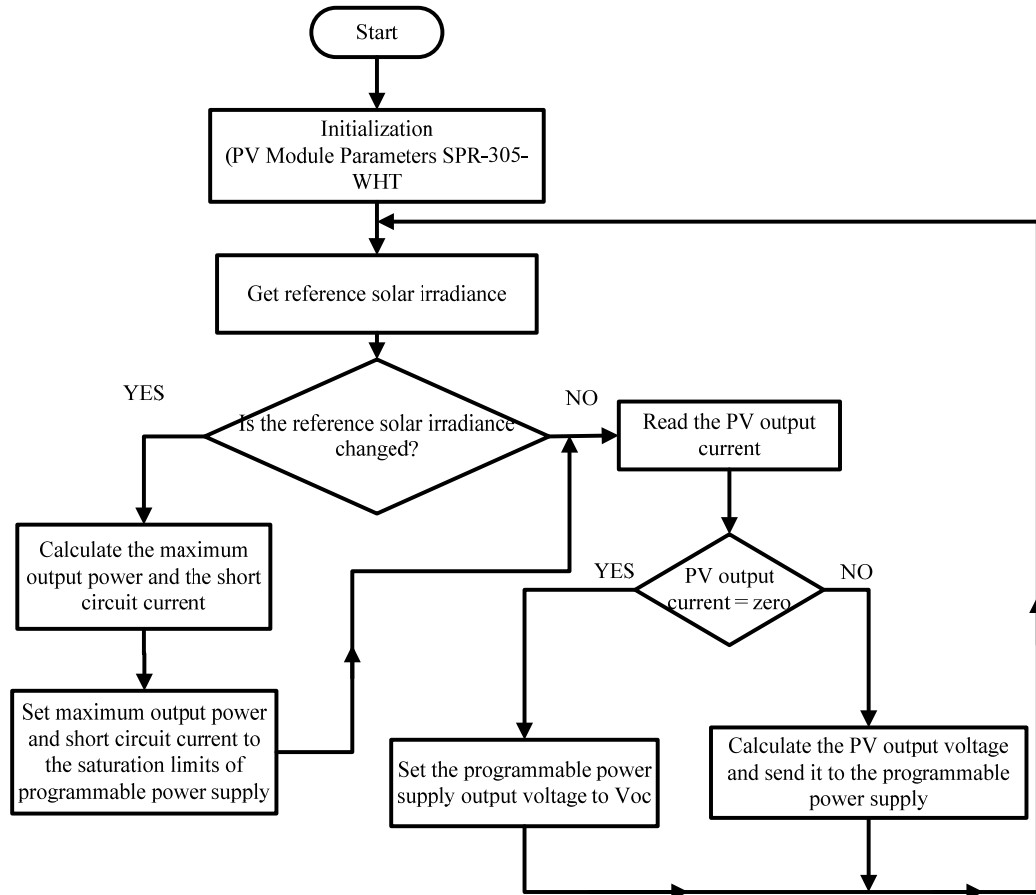

**Figure 19.** Flow chart of the microcontroller programme for PV emulator.

**Table 7.** Experimental setup parameters.

| Parameters | Value |
|---|---|
| Boost converter Switching frequency | 10 kHz |
| Boost converter inductance | 1.5 mH |
| Boost converter capacitance | 2200 µf |
| DC-bus voltage | 110 V |

**Table 8.** The PV module parameters (MODEL SUNPOWER SPR-305-WHT).

| Parameter | Value |
|---|---|
| Maximum power | 305 W at STC |
| Number of Cells | 96 |
| Current at MPP | 5.58 A |
| Voltage at MPP | 54.7 A |
| Open Circuit Voltage | 64.2 V |
| Short Circuit Current | 5.96 A |

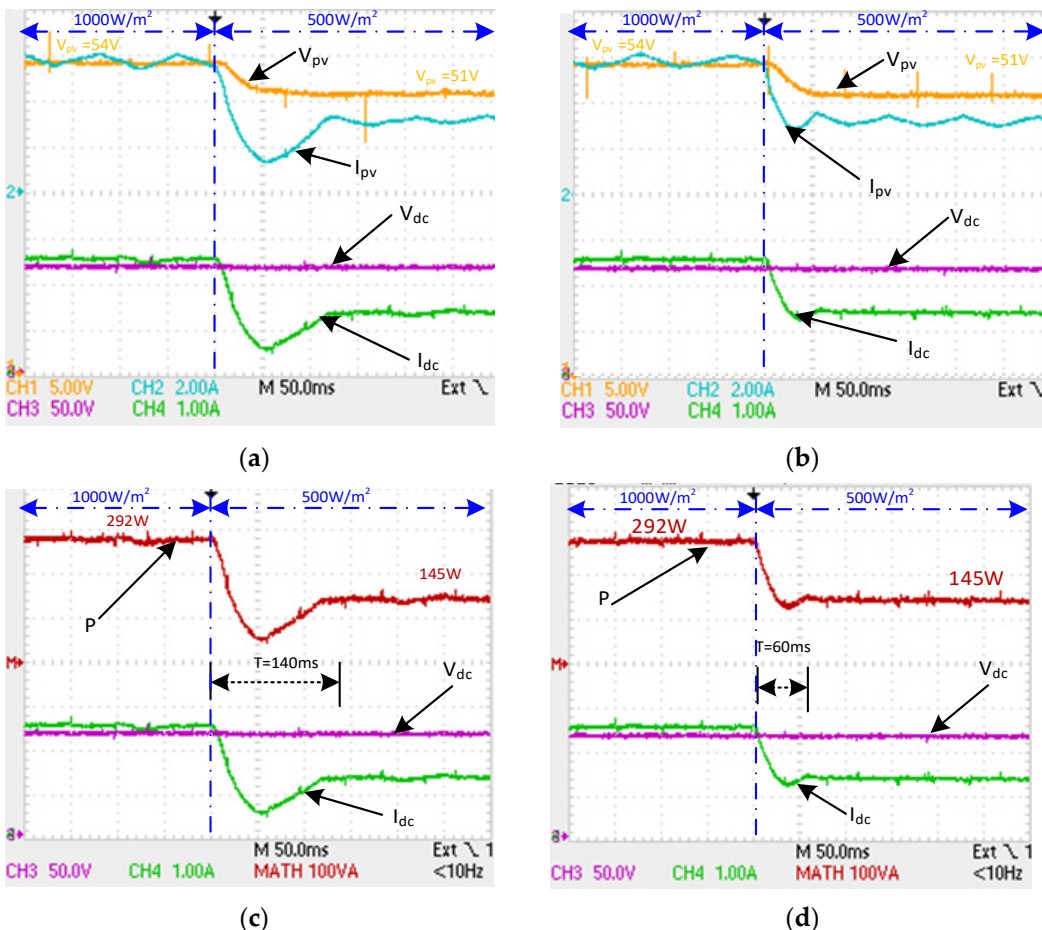

**Figure 20.** Experimental results: (**a**,**b**) The PV output voltage (VPV), PV output current (IPV), dc-bus voltage (Vdc), and dc-bus delivered current (Idc); with conventional variable step-size, and proposed one respectively, (**c**,**d**) dc-bus delivered power (P), dc-bus voltage (Vdc), and dc-bus delivered current (Idc) with conventional variable step-size and proposed one respectively.

## 7. Conclusions

In this paper, a modified variable step-size INC MPPT technique has been proposed. The drawbacks of a conventional variable step-size in INC MPP technique have been discussed in order to tackle them via the new proposed variable step-size INC technique. A new autonomous scaling factor, which is sorely dependent on PV module voltage change, has been proposed. The proposed technique is capable to enhance both the steady-state and dynamic performance response. The proposed technique is more practical in operating due to autonomous response nature under sudden changes in irradiance. Conventional, two modified and proposed variable step-size INC MPPTs have been simulated under different operating conditions using MATLAB/SIMULINK software. The feasibility and effectiveness of the proposed technique have been confirmed. The proposed MPPT tech-

nique demonstrates faster tracking speed with minimum oscillations around MPP both at steady-state and dynamic conditions. The experimental results validate the practicability and effectiveness of the new MPPT technique as well.

**Author Contributions:** I.O.-N. and K.H.A. conceived the methodology, developed the theory and simulations and wrote some sections of the manuscript. I.A. performed the experiments. M.A.E. wrote some sections of the manuscript. All authors participated in the review of the manuscript. All authors have read and agreed to the published version of the manuscript.

**Funding:** This research received no external funding.

**Conflicts of Interest:** The authors declare no conflict of interest.

## Abbreviations

The following abbreviations are used in this manuscript:

| | |
|---|---|
| D | Duty cycle |
| FOCV | Fractional open circuit voltage |
| FSCC | Fractional short circuit current |
| G | Irradiance |
| INC | Incremental conductance |
| Impp | Current at maximum power point |
| Isc | Shot circuit current |
| MPP | Maximum power point |
| MPPT | Maximum power point tracking |
| P&O | Perturb and observe |
| PV | Photovoltaic |
| P-V | Power-voltage |
| STC | Standard test conditions |
| Voc | Open circuit voltage |
| Vmpp | Voltage at maximum power point |

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
