# Peer review of "Modified Variable Step-Size Incremental Conductance MPPT Technique for Photovoltaic Systems"

_electronics, doi:10.3390/electronics10192331_

Round 1

Reviewer 1 Report

The manuscript entitled Modified Variable Step-Size Incremental Conductance MPPT 2 Technique for Photovoltaic Systems was submitted for review. In the manuscript, the Authors analyzed and developed an algorithm dedicated to MPPT. The proposed algorithm consists of two parts namely, autonomous scaling factor and slope angle variation algorithm. The autonomous scaling factor adjusts continuously the step-size without using a preset constant to control the trade-off between convergence speed and tracking precision. The authors devoted a lot of work to the preparation of the current form of the work, while the reviewer has some comments to which he asks for a response / opinion / development in the work. The order of the points is random.
Main points:
1) How was the issue of local and global MPPT solved within the proposed solution? It is about changes in the I-V characteristics as a result of (for example) local shadings.
2) What is the potential of real energy gain when the solution is applied?
3) The experience reviewer shows that the actual characteristics of PV cells are with a greater slope as shown in the figure.

How does the developed modification work for such characteristics?
4) It seems to the reviewer that the described modification of the method, consisting in changing the method of changing scalability from a constant to a variable, is an engineering rather than a scientific issue. Due to this fact, the article requires more work for tests and comparisons.
5) With which power supply / system where the PV panels simulated in the tests? Reading the EA-PS 8360 documentation, it does not seem that this power supply allows the system to be powered in accordance with the relevant standards for PV voltage (for example: EN 50530).
6) The manuscript lacks actual tests of different types of MPPT methods, showing the advantages of the proposed solution. Such solutions should be analyzed both through simulations and experimental tests.
7) What is the delay of the control system's response to parameter changes? How long does it take? How many cycles?
8) How were the parameters of converter switching frequency, converter inductance and converter capacitance selected??
9) Such tests (on a real system) presented in Figure 11 should also be performed for other methods to which the authors compare.
10) From a practical point of view, the reviewer takes part in the implementation of the autonomous PV power supply project and we use, among others, the LT8490 solution. LT8490 employs a Perturb and Observe algorithm for identifying the maximum power point. This algorithm provides accurate MPPT for slow to moderate changes in panel illumination. The panel is also scanned periodically to avoid settling on a false maximum power point for long periods of time, in the case of non-uniform panel illumination. In the solution developed by us, the oscillations mentioned by the authors do not occur. Maybe the reason is badly regulated methods that the authors refer to? Maybe the speed of the systems is too slow?
11) In the opinion of the reviewer, the work lacks a solid overview of MPPT methods. Of course, the authors took care of the new literature, but there are no review items in the literature that should be referred to. For example (or many others):
a) State of the art artificial intelligence-based MPPT techniques for mitigating partial shading effects on PV systems–A review (DOI: 10.1016/j.rser.2016.06.053)
b) Classification and comparison of maximum power point tracking techniques for photovoltaic system: A review (DOI: 10.1016/j.rser.2012.11.052)
c) Renewable and Sustainable Energy Reviews (DOI: 10.1016/j.rser.2015.07.172)
d) Maximum Power Point Tracking techniques for photovoltaic systems: A comprehensive review and comparative analysis (DOI: 10.1016/j.rser.2015.07.172 )
e) A Comparative Study on Maximum Power Point Tracking Techniques for Photovoltaic Power Systems (DOI: 10.1109/TSTE.2012.2202294)

NOTE! The reviewer is not a co-author of any of the indicated items.

Editing Notes:
1) In chapter 2, there is one sentence. The reviewer thinks this is a bad solution. You can treat subchapter 2.1 as main chapter 2, or else dispose of the place but not leave a single sentence.
2) From line 117 onwards, it seems that there is (in several places) incorrect text formatting.
3) Figure 11 is of poor quality, please correct it.

Author Response

As attached.

Reviewer 2 Report

This paper present subject related to a modification of Incremental Conductance MPPT  technique for photovoltaic systems. The advantages and limitations of different MPPT methods for PV systems was analysed. The article consists of suggestions and potentially important future research directions.  The paper is quite well prepared in editorial terms but discussed problems are not properly presented instead. In reviewer opinion the authors has not been done sufficient review of modified IC MPPT technique presented in many literature position at all. Also, the experimental results are inadequate in compare with simulation results.

Considered problem of MPPT PV systems  is up to date and very important from the point of view of modern electric power systems. I find the paper very interesting and I recommend it to be accepted after major revision.

I have few major remarks:

  1. Why the authors refer only to the standard and proposed technique experimental results, without comparing their proposal with two of other modifications described in chapter with simulation results?

  1. In reviewer's opinion authors should compare the effectiveness of the proposed modification in percentage terms with the standard solution and other modifed techniques. It will be more clarify for readers.

  1. How does changing the step of proposed method affect the losses of the converter semiconductor elements? Could authors presents some mathematical description and calculation results?

  1. In the article has not been clearly presented indication of contribution done by authors to the proposed solution.

and some minor remarks:

  1. Line 241 -literature position citation [33] in text- should be without dot before number “.[33].”
  2. In reviewer opinion the article have to small number of citations for such the wide paper
  3. Figures 3 and 5- no numerical scale (at least for power).
  4. Fig 10. - unclear description under the picture - what rig. means? 

Author Response

As attached. 

Reviewer 3 Report

Idea is clearly presented. Here are my comments:

  • In Section 3, the PV characteristic is divided into 3 different regions. If authors consider to set the left hand side into 2  regions, the right hand side should also have 2 regions. What is the division concern in the design? 
  • In Section 3.2, is the Voc estimation is written in the controller? If yes, it means appliance is required to detect irradiance and temperature data. If not, it is not a practical design.
  • The arrangement in Figure 10 needs to improve. It is hard to see each element clearly.
  • In the experimental result, the setup performance is missing.

Author Response

As attached.

Reviewer 4 Report

The paper demonstrate a new proposed modified variable step-size INC
algorithm to improve MPPT technique. The simulation and experimental results are used to validate the effectiveness of new method well. The method looks promising for industrial design and applications. Only two minor comments for your considerations are as the followings:

  1. Most figures of simulations and experiments are not in high definition. Specially the lines and fonts are not clear and sharp enough to present the multiple results in the same figures and coordinate systems.
  2. The equations are not formatted and organized well for a journal paper, such as Eqs. 1-7. Please double check the rest and all the symbols.

Author Response

The authors would like to thank the reviewers for their invaluable comments on the submitted manuscript. The main manuscript has been revised taking into account all reviewers’ comments. All the corrected, modified, and added sections are highlighted in yellow.

Reviewers' comments and reply:

1. Most figures of simulations and experiments are not in high definition. Specially the lines and fonts are not clear and sharp enough to present the multiple results in the same figures and coordinate systems.

All figures have been revised. We have also updated the fonts accordingly.

2. The equations are not formatted and organized well for a journal paper, such as Eqs. 1-7. Please double check the rest and all the symbols.

All equations have been formatted and arranged including symbols accordingly.

Round 2

Reviewer 1 Report

The manuscript entitled Modified Variable Step-Size Incremental Conductance MPPT 2 Technique for Photovoltaic Systems was submitted for review in the second revision, i.e. after the introduction of the first series of amendments. The reviewer reads the responses of the authors of the manuscript and the changes introduced. He has comments to some of the answers, which are listed below in  random order:

1) In response to the reviewer, the authors of the manuscript wrote: “… Actually, the main scope of the paper is for normal weather conditions. The proposed algorithm will work very well in partial shading however; we have to add another approach in parallel such as scanning technique…. ”. First, normal weather conditions are those that assume local shades that can introduce local peaks. Secondly, since the authors wrote that they will use an additional algorithm working in parallel to scan the characteristics, they should present such results in the article.

2) The authors of the manuscript wrote: “… The performance indicators point to the fact that there is a real energy gain when the proposed MPPT is applied in PV system. This is due to its restricted search range; the algorithm uses short time to reach MPP to save Energy ... ", this indicator is presented in Table 4. It should be emphasized that if the solution is not resistant to local maximum (and after finding a local maximum it will not look for a global maximum), a situation may arise in the characteristics of which will be more complex (example in the figure below) and the algorithm will find a local, not a global maximum. In that case, it will be quite a waste of gaining energy.

3) How was the accuracy of MPP tracking determined, since there is no local maximum resistance and the authors do not have a PV cell simulator, but an programmable power supply??

4) Since the authors use only a programmable power supply, without implemented PV characteristics (in accordance with the standards), it is necessary to justify the correctness of the solution's operation.

5) There is no in-depth comparison with other MPPT methods. A fair comparison.

6) The authors wrote: „…Unfortunately, authors were unable to re-run the experimental setup due to limited access to the laboratory during these difficult times of Covid-19 worldwide pandemic…” If the authors did not have enough time, then you should wait and perform a complete set of tests, analyzes and studies, and not send a partially prepared article for review.

7) The authors wrote: „… We have considered your suggestion and have included the above listed technical papers to improve the general overview of the manuscript.…” .  The reviewer did not mean to add the indicated publications by the authors (NOTE! In any of the publications indicated, the reviewer is not the author), but to conduct an analysis and compare other MPPT methods with the proposed one.

8) The I-V characteristic (for PV) used is not actually present in modern panels. The reviewer has previously pointed out this. Below, the reviewer lists the PV characteristics used in the first revision and in the present one. How do these characteristics differ from each other?

9) Since the authors of the manuscript only have an ordinary programmable power supply at their disposal, how were they able to determine the time to reach the maximum power point, which is 12.6 ms (Table 4) ?

10) To article suitable for publication, comparative analysis with other methods should be carried out much more carefully. Analyzes should be performed for various irradiation conditions, for dynamically changing conditions. The lack of appropriate equipment is not an argument as any new method must be sufficiently confirmed.
Please check how the analysis of the new solution was carried out, for example in the article:
-Effect of Various Incremental Conductance MPPT Methods on the Charging of Battery Load Feed by Solar Panel, DOI: 10.1109/ACCESS.2021.3091502, (In the solar photovoltaic system, the variable step size selection method for INC is proposed and compared);

- A new MPPT design using variable step size perturb and observe method for PV system under partially shaded conditions by modified shuffled frog leaping algorithm- SMC controller, DOI: 10.1016/j.seta.2021.101056.

NOTE! In any of the works cited, the reviewer is not the author.

The manuscript submitted for review requires a large amount of additional work that could be suitable for publication.

Author Response

As attached.

Reviewer 2 Report

The authors have complied with all the recommendations of the reviewer. I recommend that the article can be published

Author Response

Thank you.

Round 3

Reviewer 1 Report

A manuscript entitled Modified Variable Step-Size Incremental Conductance MPPT Technique for Photovoltaic Systems was submitted for review, containing significant improvements to the previous version, including all comments indicated in the review. The reviewer appreciates the work put into it, and the time spent on carrying out additional tests and simulations. The reviewer is satisfied with all responses and believes that the scientific value of the article has been significantly improved. The reviewer has one practical question and two editorial requests. After considering these questions, the reviewer does not plan to submit any further comments.

Practical aspect:

1) The reviewer did not find any information in the manuscript about the actual implementation of the tests and simulations in terms of whether the tested system is connected to the grid (on-grid type) or to an external load (off-grid type). Of course, the reviewer understands that this is not the basis of the article, but such information should be included in the description in the section on simulations and tests. The experimental model is shown in figure 12, but the "load" part is not marked there.

Editing aspects:

1) Please align the numbering of the formulas in the places where they are defined (as required by the journal),

2) Please unification descriptions of tables, in some cases at the end of the description of a dot on some, of its (dots) none - for example table 2, 5, 6, 7.

Author Response

The authors would like to thanks Editor-in-Chief, Associate Editor and reviewers for their invaluable comments on the submitted manuscript.

Manuscript No. 1351832

The main manuscript has been revised taking into account all reviewers’ comments. All the corrected/modified/added sections are highlighted in yellow.

Reviewer: 1
1) The reviewer did not find any information in the manuscript about the actual implementation of the tests and simulations in terms of whether the tested system is connected to the grid (on-grid type) or to an external load (off-grid type). Of course, the reviewer understands that this is not the basis of the article, but such information should be included in the description in the section on simulations and tests. The experimental model is shown in figure 12, but the "load" part is not marked there.

Response

The authors would like to thank the reviewer again for his efforts and feedback on the manuscript. The tested PV system is grid connected system where the battery represents the ac grid and interfacing inverter. It has been highlighted clearly in the revised manuscript.

Editing aspects:

1) Please align the numbering of the formulas in the places where they are defined (as required by the journal),

Response

Thank you for your comment, we have aligned the numbering of the formulas.

2) Please unification descriptions of tables, in some cases at the end of the description of a dot on some, of its (dots) none - for example table 2, 5, 6, 7.

Response

Thank you for your comment. We have corrected all Tables including Tables 2, 5, 6, and 7.

Round 4

Reviewer 1 Report

Well done.